# Replacing Histogram with Smooth Empirical Probability Density Function Estimated by K-Moments

Demetris Koutsoyiannis

School of Civil Engineering, National Technical University of Athens, Heroon Polytechneiou 5,
GR 157 80 Zographou, Greece; dk@itia.ntua.gr

**Abstract:** Whilst several methods exist to provide sample estimates of the probability distribution function at several points, for the probability density of continuous stochastic variables, only a gross representation through the histogram is typically used. It is shown that the newly introduced concept of knowable moments (K-moments) can provide smooth empirical representations of the distribution function, which in turn can yield point and interval estimates of the density function at a large number of points or even at any arbitrary point within the range of the available observations. The proposed framework is simple to apply and is illustrated with several applications for a variety of distribution functions.

**Keywords:** statistics; stochastics; distribution function; probability density; statistical estimation; histogram; knowable moments

## 1. Introduction

The concepts of *distribution function* ($F(x)$) and *probability density* ($f(x)$) of a *stochastic* (or *random*) *variable* ($\underline{x}$ with realizations $x$; notice the notational convention to underline stochastic variables—the Dutch convention) are central in Kolmogorov's foundation of probability in 1933 [1,2] and its applications. (Here, we adhere to the original Kolmogorov's terms, noting that in the English literature $F(x)$ is also known as the cumulative distribution function.) Their estimation, based on a sample of $\underline{x}$, is crucial in most applications.

The standard estimate of $F(x)$ was again introduced by Kolmogorov [3], who termed it the *empirical distribution*, and is still in common use [4], sometimes under the name *sample distribution function*. Denoting $\underline{x}_{(i:n)}$ the $i$th *order statistic* in a sample of size $n$, i.e., the $i$th smallest of the $n$ variables arranged in increasing order ($\underline{x}_{(1:n)} \leq \underline{x}_{(2:n)} \leq \ldots \leq \underline{x}_{(n:n)}$), the standard estimate is:

$$\hat{F}(x) = \frac{n_x}{n}, \quad x_{(n_x-1:n)} < x \leq x_{(n_x:n)} \tag{1}$$

where $n_x$ is the number of the values in the observed sample that do not exceed $x$ (an integer). The function $\hat{F}(x)$ has a staircase form with discontinuities at each of the observations $x_i$ (see Appendix A for an illustration and more details). If $\underline{x}$ is a continuous variable, almost certainly the values $x_i$ are distinct and the jump of $\hat{F}(x)$ at $x_i$ equals $1/n$. Apparently then, the representation of the continuous $F(x)$ by the discontinuous $\hat{F}(x)$ is not ideal. This representation has two additional problems, i.e., (a) for $x < \underline{x}_{(1:n)}$ the resulting $\hat{F}(x)$ is 0, and (b) for $x \geq \underline{x}_{(n:n)}$ the resulting $\hat{F}(x)$ is 1. Both these are problematic results if the variable $\underline{x}$ is unbounded.

Both the latter problems are usually tackled with the notion of so-called *plotting positions* (e.g., [5])—a rather unsatisfactory name. There is a variety of plotting position formulae, each of which is a modification of Equation (1) by adding one or two constants in the numerator and/or the denominator. A review of these formulae and their justification is provided by Koutsoyiannis (2022) [6], along with a set of new proposed ones, derived using the theoretical properties of order statistics.

One could also think to tackle the discontinuity problem by replacing the staircase function by some type of interpolation (e.g., linear or logarithmic). However, the thus resulting function would again be too rough for practical applications. In particular, a rough function, where the roughness is a statistical sampling effect rather than an intrinsic property of the distribution function, cannot support reliable estimation of derivatives. Since the density $f(x)$ is the first derivative of the distribution $F(x)$, the above framework cannot be used to estimate the former. An illustration of the roughness of $F(x)$ and $f(x)$, if the former is estimated from Equation (1) and the latter by the numerical derivative of the former, is provided in Appendix A (in particular, in Figure A1).

Methods for a detailed estimation of the probability density at different points $x$ are lacking. Instead, a gross estimation based on the histogram constitutes the standard representation of the density. Yet, the estimation of the density is necessary for several tasks, e.g., those involving hazard (where the hazard function is defined as $f(x)/(1 - F(x))$; see [7]) or entropy (where entropy is the expectation of $-\ln f(\underline{x})$ properly standardized; see Section 3.3).

The histogram representation is constructed by choosing (a) an interval $[a, b]$ that contains all observations (often called the range) and (b) a number $k$ of equally spaced bins, so that the width of the bins is

$$w = \frac{b - a}{k} \qquad (2)$$

The number $k$ is usually small, typically chosen by the old Sturges' rule [8]:

$$k = \log_2(2n) = 1 + \frac{\ln n}{\ln 2} \qquad (3)$$

For example, for a sample size $n = 100$, this results in $k \approx 8$. The underlying rationale for the rule and comparisons with additional rules are provided by Scott [9]. Once $a, b, k$ are chosen, the density estimate is

$$\hat{f}(x) = \frac{n_{x_{i+1}} - n_{x_i}}{n\,w}, \quad i = 1, \dots, k, \quad n_{x_i} < x \le n_{x_{i+1}}, \quad x_i = a + (i - 1)w \qquad (4)$$

The entire framework for the construction of histogram entails subjectivity, and lacks detail and accuracy.

As an alternative to constructing a smooth empirical density function, Rosenblatt [10] and Parzen [7] proposed the use of kernel smoothing. They have been followed by several researchers who developed the method further [11–14]. This is regarded as a non-parametric method, but it uses a specified kernel function which contains parameters, where both the function expression and its parameters are arbitrarily chosen by the user. Typical kernel functions include uniform, triangular, quadratic (Epanechnikov), biweight, triweight, normal, and even atomic kernels [15,16]. While the methods of this type may provide a smooth function, their reliability is questionable, and the results are affected by a great deal of subjectivity owing to user choices. These characteristics are illustrated in Appendix A (in particular, in Figure A2).

Here, we propose a new method of estimation of the probability density based on the concept of *knowable moments*, abbreviated as *K-moments*. As shown in [6] (pp. 147–249) and summarized in the next section, the noncentral K-moment of order $p$, $K'_p$, can: (a) yield a reliable and unbiased estimate $\hat{K}'_p$ from a sample, up to order $p$ equal to the sample size $n$, and (b) be assigned a value of return period, or equivalently distribution function estimate, $\hat{F}\left(\hat{K}'_p\right)$. In addition, the sequence of estimates $\hat{F}\left(\hat{K}'_p\right)$ form a smooth function which can be used in estimating the probability density at a large number of points (as opposed to the roughness implied by Equation (1)—or other versions thereof based on plotting positions and eventually on order statistics—and as illustrated in Figure A1 of Appendix A). The estimate $\hat{K}'_p$ is based on the notion of order statistics but, contrary to their standard use, where only one sample value is used at a time, the K-moment framework combines many

order statistics simultaneously, thus converting a rough arrangement to smooth, without involving any subjectively chosen kernel function.

## 2. K-Moments and Their Relevance

### 2.1. Definition and Interpretation

We recall that the *noncentral* (or *raw*) *moment* of order $p$ of a stochastic variable $\underline{x}$ is defined as the expectation:

$$\mu'_p := \mathrm{E}[\underline{x}^p] = \int\limits_{-\infty}^{\infty} x^p f(x)\mathrm{d}x \tag{5}$$

with $\mu'_1 = \mathrm{E}[\underline{x}] =: \mu$ representing the mean. Let $\underline{x}_1, \underline{x}_2, \ldots, \underline{x}_p$ be independent copies of $\underline{x}$, forming a sample. Then, the standard estimator of $\mu'_p$ from the sample is

$$\underline{\hat{\mu}}'_p = \frac{1}{n}\sum_{i=1}^{n} \underline{x}_i^p \tag{6}$$

It is well known that the estimator of the noncentral moment is unbiased, i.e.,

$$\mathrm{E}\left[\underline{\hat{\mu}}'_p\right] = \mu'_p \tag{7}$$

While unbiasedness is theoretically guaranteed, the convergence of $\hat{\mu}'_p$ to $\mu'_p$ is extraordinarily slow if $p$ is not very low [17]. In practice, for large or moderate $p$ (greater than 2 or 3, depending on the sample size), what we actually calculate by applying the standard estimator, is an estimate of some extreme quantity, rather than an estimate of the moment $\mu'_p$. To see this, we recall that for positive $x_i$ and for large (or even modest) $p$ the approximation $\sum\limits_{i=1}^{n} x_i^p \approx \left(\max\limits_{1\leq i\leq n}(x_i)\right)^p$ holds, so that

$$\hat{\mu}'_p = \frac{1}{n}\sum_{i=1}^{n} x_i^p \approx \frac{1}{n}\left(\max_{1\leq i\leq n}(x_i)\right)^p \tag{8}$$

Thus, unless $p$ is very small, we cannot infer the value of $\mu'_p$ from data, i.e., it is *unknowable* (Koutsoyiannis, 2019 [18]; see also [6], pp. 125–126). This is the case even if the sample size $n$ is extraordinarily large [6,18]. This reduces the power of the concept of moments for statistical inference, which is the formal probabilistic induction (the modern version of the Aristotelian epagoge/ἐπαγωγή) and is based on expectations, estimated from samples by virtue of stationarity and ergodicity.

To obtain a *knowable* moment of order $p$, Koutsoyiannis [18] raised $x$ to a low power, $q < p$ and for the remaining $(p-q)$ multiplicative terms, replaced $x$ with $F(x)$, hence defining the *noncentral knowable moment* (or *noncentral K-moment*) of orders $(p, q)$ as:

$$K'_{pq} := (p-q+1)\mathrm{E}\left[(F(\underline{x}))^{p-q}\underline{x}^q\right], \quad p \geq q \tag{9}$$

with the most interesting special case obtained for $q = 1$:

$$K'_p := p\mathrm{E}\left[(F(\underline{x}))^{p-1}\underline{x}\right], \quad p \geq 1 \tag{10}$$

Koutsoyiannis [6,18] also introduced other types of K-moments, among which here we will use, in addition to $K'_p$, the *tail-based (noncentral) moments*:

$$\overline{K}'_p := p\mathrm{E}\left[(1 - F(\underline{x}))^{p-1}\underline{x}\right] = p\mathrm{E}\left[(\overline{F}(\underline{x}))^{p-1}\underline{x}\right], \quad p \geq 1 \tag{11}$$

where $\overline{F}(\underline{x}) := 1 - F(\underline{x})$ is the tail function.

For an interpretation of K-moments we consider the maximum of $p$ stochastic variables $\underline{x}_1, \underline{x}_2, \ldots, \underline{x}_p$, which is the largest ($p$th) order statistic:

$$\underline{x}_{(p)} := \underline{x}_{(p:p)} = \max\left(\underline{x}_1, \underline{x}_2, \ldots, \underline{x}_p\right) \tag{12}$$

It is readily obtained that if $F(x)$ is the distribution function of $\underline{x}$ and $f(x)$ its probability density function, then those of $\underline{x}_{(p)}$ are:

$$F^{(p)}(x) = (F(x))^p, \quad f^{(p)}(x) = pf(x)(F(x))^{p-1} \tag{13}$$

where the former is the product of $p$ instances of $F(x)$ (justified by the fact that the variables $\underline{x}_i$ are independent copies of $\underline{x}$, by definition of the sample concept), while the latter is the derivative of $F^{(p)}(x)$ with respect to $x$. The *expected maximum of order $p$* of $\underline{x}$, i.e., the expected value of $\underline{x}_{(p)}$, is therefore:

$$\mathrm{E}\left[\underline{x}_{(p)}\right] = \mathrm{E}\left[\max\left(\underline{x}_1, \underline{x}_2, \ldots, \underline{x}_p\right)\right] = p\mathrm{E}\left[(F(\underline{x}))^{p-1}\underline{x}\right] \tag{14}$$

and this is precisely the noncentral K-moment $K'_p$. Likewise, the minimum of the $p$ variables

$$\underline{x}_{(1:p)} = \min\left(\underline{x}_1, \underline{x}_2, \ldots, \underline{x}_p\right) \tag{15}$$

has expectation:

$$\mathrm{E}\left[\min\left(\underline{x}_1, \underline{x}_2, \ldots, \underline{x}_p\right)\right] = p\mathrm{E}\left[(1 - F(\underline{x}))^{p-1}\underline{x}\right] = p\mathrm{E}\left[(\overline{F}(\underline{x}))^{p-1}\underline{x}\right] \tag{16}$$

which is the tail K-moment $\overline{K}'_p$.

It is thus easy to see that the sequence of $K'_p$ is non-decreasing and that of $\overline{K}'_p$ is non-increasing as $p$ increases.

### 2.2. Estimation of K-Moments

Several estimators of the K-moments have been developed in [6], among which here we use the unbiased ones [6] (pp. 193–196 and 229–231):

$$\hat{K}'_p = \sum_{i=1}^{n} b_{inp} \, \underline{x}_{(i:n)}, \quad \hat{\overline{K}}'_p = \sum_{i=1}^{n} b_{inp} \, \underline{x}_{(n-i+1:n)} = \sum_{j=1}^{n} b_{n-j+1,n,p} \, \underline{x}_{(j:n)} \tag{17}$$

where

$$b_{inp} := \begin{cases} 0, & i < p \\ \dfrac{p}{n} \dfrac{\Gamma(n-p+1)}{\Gamma(n)} \dfrac{\Gamma(i)}{\Gamma(i-p+1)}, & i \geq p > 0 \end{cases} \tag{18}$$

and $\Gamma(a) := \int\limits_{0}^{\infty} t^{a-1}\mathrm{e}^{-t}\mathrm{d}t$ is the gamma function. This allows estimation of the K-moments for any order $p$ from 1 to $n$. Thus, from a sample of size $n$ we can estimate a number of noncentral and tail moments equal to $2n - 1$ (notice that $\hat{K}'_1 = \hat{\overline{K}}'_1 = \hat{\mu}$), with high reliability and low uncertainty (see [6], p. 195). It can be easily verified that for any order $p$,

$$\sum_{i=1}^{n} b_{inp} = 1 \tag{19}$$

which is a necessary condition for unbiasedness. Special cases of K-moment estimator coefficients $b_{inp}$ are shown in Table 1.

**Table 1.** Special cases of K-moment estimator coefficients (adapted from [6], p. 194).

| Case | $b_{inp}$ | Case | $b_{inp}$ |
|---|---|---|---|
| $p = 1$ | $b_{in1} = \frac{1}{n}$ | $p = n - 1$ | $b_{n-1,n,n-1} = \frac{1}{n}, b_{n,n,n-1} = 1 - \frac{1}{n}$ |
| $p = 2$ | $b_{in2} = \frac{2}{n} \frac{i-1}{n-1}$ | $p = n$ | $b_{nnn} = 1$ |
| $p = 3$ | $b_{in2} = \frac{3}{n} \frac{i-1}{n-1} \frac{i-2}{n-2}$ | $i = n$ | $b_{nnp} = \frac{p}{n}$ |
| $p = 4$ | $b_{in4} = \frac{4}{n} \frac{i-1}{n-1} \frac{i-2}{n-2} \frac{i-3}{n-3}$ | $i = p$ | $b_{pnp} = p\mathrm{B}(p, n - p + 1)$ *  symmetry: $b_{pnp} = b_{n-p,n,n-p}$  (minimum at $p = n/2$) |

* $\mathrm{B}(a, b)$ is the beta function.

The fact that $b_{inp} = 0$ for $i < p$ suggests that, as the moment order increases, progressively fewer data values determine the moment estimate, until it remains only one (the maximum for the noncentral moment and the minimum for the tail moment), when $p = n$, with $b_{nnn} = 1$. Furthermore, if $p > n$ then $b_{inp} = 0$ for all $i$, $1 \leq i \leq n$, and therefore estimation becomes impossible.

*2.3. Estimation of the Distribution Function at K-Moment Values*

Order statistics have an important advantage over other statistics, as to each of them we can assign a value of the distribution function, or equivalently, the return period. We recall that for a specific event $A$ the return period, $T$, is defined to be the mean time between consecutive occurrences of the event $A$. Assuming that the event is the exceedance of a certain level $x$, i.e., $A := \{\underline{x} > x\}$, the return period is related to the distribution function by

$$\frac{T}{D} = \frac{1}{\overline{F}(x)} = \frac{1}{1 - F(x)} \tag{20}$$

where $D$ is a time window width (time scale or time step) used to define the metric $x$ (e.g., $D = 1$ year if $x$ is the annual rainfall total). This is known as the return period of maxima.

Likewise, for the non-exceedance of the value $x$, i.e., $A := \{\underline{x} \leq x\}$, the return period (of minima) is

$$\frac{\overline{T}}{D} = \frac{1}{F(x)} \tag{21}$$

As seen above, the K-moments are closely related to order statistics and therefore it becomes possible to assign return periods to K-moment values. Intuitively, we can expect that the $p$th noncentral K-moment (the value $x = K'_p$) will correspond to a return period of about $2pD$. This is precise for a symmetric distribution and for $p = 1$, as $K'_1$ is the mean value which has return period $2D$. (For instance, the mean of annual rainfall is exceeded, or non-exceeded, on the average, every two years). As we will see below, the return period cannot be much lower than $2pD$ for any $p$ and for any distribution.

Generally, we can expect that the return period is an increasing function of the moment order $p$, with a relationship of approximate proportionality:

$$\frac{T\left(K'_p\right)}{D} \underset{\sim}{\propto} p \tag{22}$$

As justified above, a coefficient of proportionality equal to 2 can be used as a first rough approximation (rule of thumb that helps intuition). However, more precisely, the coefficient of proportionality, call it $\Lambda_p$, depends on the distribution function and the order $p$, but its variation is not wide. The precise definition of $\Lambda_p$ is:

$$\Lambda_p := \frac{1}{p\left(1 - F\left(K'_p\right)\right)} \tag{23}$$

For given $p$ and distribution function $F(x)$, $K'_p$, $T\left(K'_p\right)$, $F\left(K'_p\right)$ and $\Lambda_p$ are analytically or numerically determined from their definitions, but this might be complicated. However, the small variation of $\Lambda_p$ with $p$ makes possible a very good approximation if we first accurately determine (a) the value $\Lambda_1$ for $p = 1$, and (b) the asymptotic value $\Lambda_\infty$. The value $\Lambda_1$ is very easy to determine, as it refers to the return period of the mean:

$$\Lambda_1 = \frac{1}{1 - F(\mu)} \tag{24}$$

and can also be reliably estimated from a sample by Equation (1).

Furthermore, in a number of customary distributions, specifically those belonging to the domain of attraction of the Extreme Value Type I distribution (see [6] (pp. 76–79 and 235–236), $\Lambda_\infty$ has a constant value, independent of the distribution:

$$\Lambda_\infty = e^\gamma = 1.781 \tag{25}$$

where $\gamma$ is the Euler constant. For heavy tailed distributions $\Lambda_\infty$ depends on the higher tail index only. Details are given in [6] (pp. 208–215), while for the distributions examined in this study, which are listed along with their main characteristics in Table 2, they are shown in Table 3.

**Table 2.** Main characteristics of the distribution functions used in the illustrations.

| Name, Parameters *, Domain | Probability Density Function, f(x) | Mean, $\mu'_1$ | Variance, $\mu_2$ | Entropy $\Phi$ |
|---|---|---|---|---|
| Exponential $\mu > 0$, $x \geq 0$ | $e^{-x/\mu}/\mu$ | $\mu$ | $\mu^2$ | $\ln e\mu$ |
| Normal $\mu \in \mathbb{R}$, $\sigma > 0$, $x \in \mathbb{R}$ | $\dfrac{\exp\left(-\frac{(x-\mu)^2}{2\sigma^2}\right)}{\sqrt{2\pi}\sigma}$ | $\mu$ | $\sigma^2$ | $\ln\left(\sqrt{2e\pi}\,\sigma\right)$ |
| Lognormal $\varsigma > 0$, $\lambda > 0$, $x \geq 0$ | $\dfrac{\exp\left(-\frac{1}{2\varsigma^2}\left(\ln\left(\frac{x}{\lambda}\right)\right)^2\right)}{\sqrt{2\pi}\,\varsigma x}$ | $e^{\frac{\varsigma^2}{2}}\lambda$ | $e^{\varsigma^2}\left(e^{\varsigma^2}-1\right)\lambda^2$ | $\ln\left(\sqrt{2e\pi}\,\lambda\varsigma\right)$ |
| Pareto $\xi > 0$, $\lambda > 0$, $x \geq 0$ | $\frac{1}{\lambda}\left(1 + \xi\frac{x}{\lambda}\right)^{-1-\frac{1}{\xi}}$ | $\frac{\lambda}{1-\xi}$ | $\frac{\lambda^2}{(1-\xi)^2(1-2\xi)}$ | $\xi + \ln e\lambda$ |

* Parameter notation: $\mu$ mean; $\sigma$ standard deviation; $\lambda$ scale parameter, $\xi$ upper tail index; $\varsigma$ shape parameter other than tail index.

**Table 3.** Characteristic $\Lambda$ values for the distributions of Table 2.

| Distribution | $\Lambda_1$ | $\Lambda_\infty$ | $\overline{\Lambda}_1$ | $\overline{\Lambda}_\infty$ |
|---|---|---|---|---|
| Exponential | $e = 2.718$ | $e^\gamma = 1.781$ * | $\frac{e}{e-1} = 1.582$ | $1$ |
| Normal | $2$ | $e^\gamma = 1.781$ | $2$ | $e^\gamma = 1.781$ |
| Lognormal | $\frac{2}{\mathrm{erfc}(\varsigma/2^{3/2})}$ | $e^\gamma = 1.781$ | $\frac{2}{2-\mathrm{erfc}(\varsigma/2^{3/2})}$ | $1$ [†] |
| Pareto | $(1-\xi)^{-1/\xi}$ | $\Gamma(1-\xi)^{1/\xi}$ | $\frac{1}{1-(1-\xi)^{1/\xi}}$ | $1$ |

* $\gamma = 0.577$ is the Euler's constant. [†] The theoretically consistent value is $e^\gamma$, but the convergence to the limit is very slow and thus the value 1 (like in the exponential and Pareto distribution) provides more accurate numerical results for typical sample sizes.

Given $\Lambda_1$ and $\Lambda_\infty$, the coefficient $\Lambda_p$ for any order $p$ can be satisfactorily (see [6], (pp. 216–218)) approximated with the following simple relationship:

$$\Lambda_p \approx \Lambda_\infty + \frac{(\Lambda_1 - \Lambda_\infty)}{p} \tag{26}$$

while more accurate approximations are given in [6] (pp. 208–215) and also discussed in Section 4 below. Equation (26) yields a linear relationship between the return period $T$ and $p$:

$$\frac{T\left(K'_p\right)}{D} = p\Lambda_p \approx \Lambda_\infty p + (\Lambda_1 - \Lambda_\infty) \tag{27}$$

from which we find

$$F\left(K'_p\right) = 1 - \frac{1}{\Lambda_\infty p + \Lambda_1 - \Lambda_\infty} \tag{28}$$

Likewise, for the tail moments we have:

$$\overline{\Lambda}_p := \frac{1}{p\, F\left(\overline{K}'_p\right)} \tag{29}$$

with

$$\overline{\Lambda}_1 = \frac{1}{F(\mu)} = \frac{\Lambda_1}{\Lambda_1 - 1} \tag{30}$$

while the limiting value $\overline{\Lambda}_\infty$ depends only on the lower tail index and is $\overline{\Lambda}_\infty = \Lambda_\infty = e^\gamma$ for the normal and other symmetric distributions, and $\overline{\Lambda}_\infty = 1$ for distributions with lower-tail index equal to one such as the exponential and Pareto distributions (see Table 3).

Again we can use the approximation:

$$\overline{\Lambda}_p \approx \overline{\Lambda}_\infty + \frac{\overline{\Lambda}_1 - \overline{\Lambda}_\infty}{p} \tag{31}$$

and find

$$F\left(\overline{K}'_p\right) = \frac{1}{\overline{\Lambda}_\infty p + \left(\overline{\Lambda}_1 - \overline{\Lambda}_\infty\right)} \tag{32}$$

## 3. Results

### 3.1. Estimation of Probability Density

Based on the above framework, we can estimate $2n - 1$ values of K-moments $\hat{K}'_p$ and $\hat{\overline{K}}'_p$, namely the ordered values $\hat{\overline{K}}'_n \leq \hat{\overline{K}}'_{n-1} \leq \ldots \leq \hat{\overline{K}}'_1 \equiv \hat{K}'_1 \leq \hat{K}'_2 \leq \ldots \hat{K}'_{n-1} \leq \hat{K}'_n$. To each one of them we can assign an empirical estimate of the distribution function $\hat{F}\left(\hat{K}'_p\right)$ and $\hat{F}\left(\hat{\overline{K}}'_p\right)$. This is similar to the distribution function values assigned via the order statistics (the plotting positions) except that (a) the number of values in the K-moments framework is twice that of the case of order statistics, and (b) the arrangement of point estimates in the former case is smooth, while in the latter is rough. The smooth arrangement allows a direct estimate of the probability density for $1 \leq p \leq n - 1$. As at points $x = \hat{K}'_p$ and $x = \hat{\overline{K}}'_p$, the values of the distribution function have been estimated from Equations (28) and (32), i.e., $F(x) = \hat{F}\left(\hat{K}'_p\right)$ and $F(x) = \hat{F}\left(\hat{\overline{K}}'_p\right)$, it is then straightforward to approximate the derivative $f(x) = \mathrm{d}F(x)/\mathrm{d}x$ by its discrete version:

$$\hat{f}(x) = \begin{cases} \dfrac{\hat{F}\left(\hat{K}'_{p+1}\right) - \hat{F}\left(\hat{K}'_p\right)}{\hat{K}'_{p+1} - \hat{K}'_p}, & \hat{K}'_p \leq x < \hat{K}'_{p+1} \\[3ex] \dfrac{\hat{F}\left(\hat{\overline{K}}'_p\right) - \hat{F}\left(\hat{\overline{K}}'_{p+1}\right)}{\hat{\overline{K}}'_p - \hat{\overline{K}}'_{p+1}}, & \hat{\overline{K}}'_{p+1} \leq x < \hat{\overline{K}}'_p \end{cases} \tag{33}$$

This procedure will result in $2n - 2$ different values of the density $\hat{f}(x)$. In Section 4 we will see that it is possible to expand the number of estimation points by using non-integer orders $p$, but in general the number $2n - 2$ is more than enough. This is illustrated in Figure 1 for a synthetic sample of size 100, generated from a lognormal distribution, also in comparison with a histogram of $k = 10$ bins (slightly more than the number resulting from Sturges' rule, $k \approx 8$).

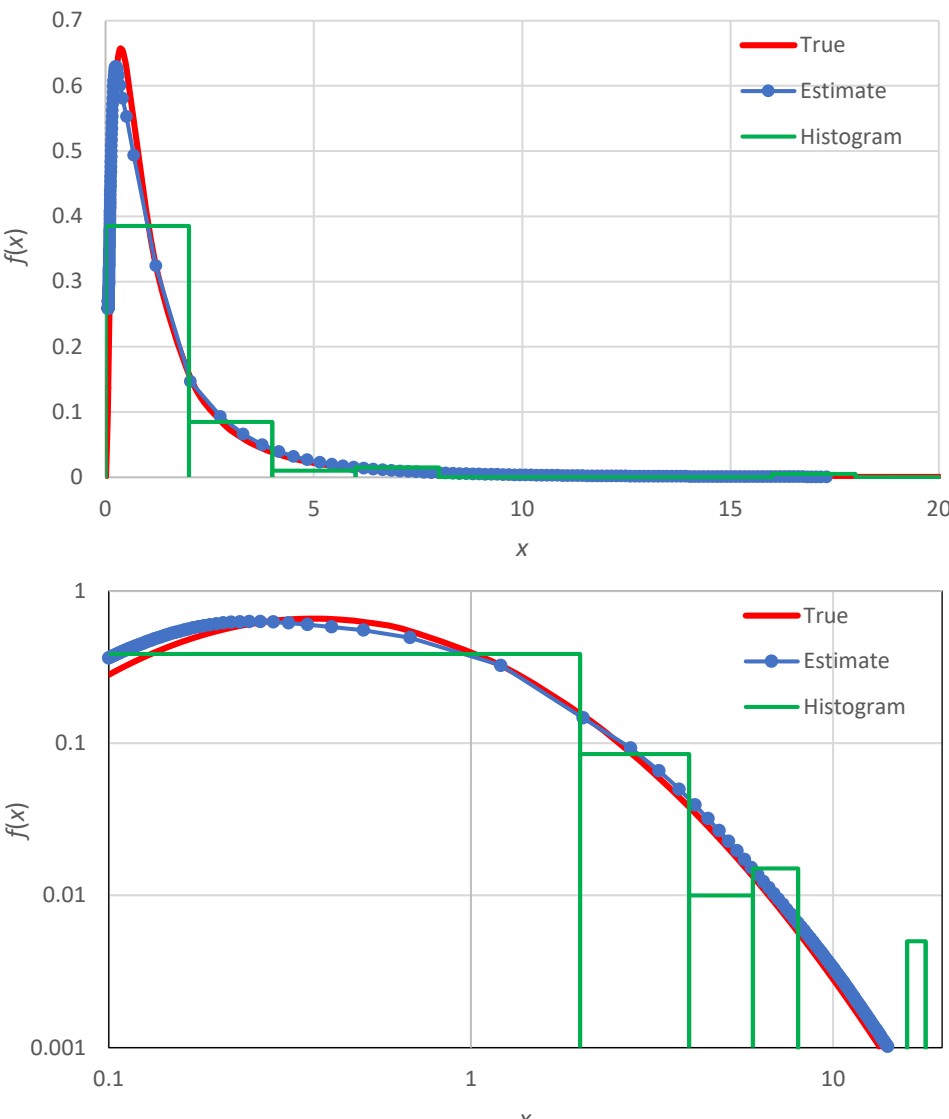

**Figure 1.** Illustration of the probability density estimate using the proposed method, plotted in Cartesian (**upper**) and logarithmic (**lower**) axes. A data series of $n = 100$ values was used, generated from a lognormal distribution (see Table 2) with parameters $\varsigma = \lambda = 1$. The density of the generating distribution is marked as "true". The points marked as "estimate" are calculated by the proposed method (Equation (33)) and their abscissae are the midpoints of the intervals $\left( \hat{K}'_p, \hat{K}'_{p+1} \right)$ and $\left( \hat{\bar{K}}'_{p+1}, \hat{\bar{K}}'_p \right)$. For comparison, the histogram of 10 bins, calculated as in Equation (4) for the range [0, 20] (width $w = 2$) is also shown.

Clearly the histogram representation of the true density (also shown in in Figure 1) is poor: five of its ten bins are empty, its shape is rough and the increasing limb of the density (for small $x$) is not captured at all. In contrast, the proposed method results in a very faithful representation of the true density.

*3.2. Uncertainty Assessment*

A more systematic illustration is made by means of Monte Carlo distribution with 100 realizations of samples of 100 items each, for all four distributions listed in Table 2. The Monte Carlo simulations allow assessing the estimation uncertainty in terms of prediction limits. The results of this investigation are shown in Figure 2 for the exponential distribution, Figure 3 for the normal distribution, Figure 4 for the lognormal distribution and Figure 5 for the Pareto distribution.

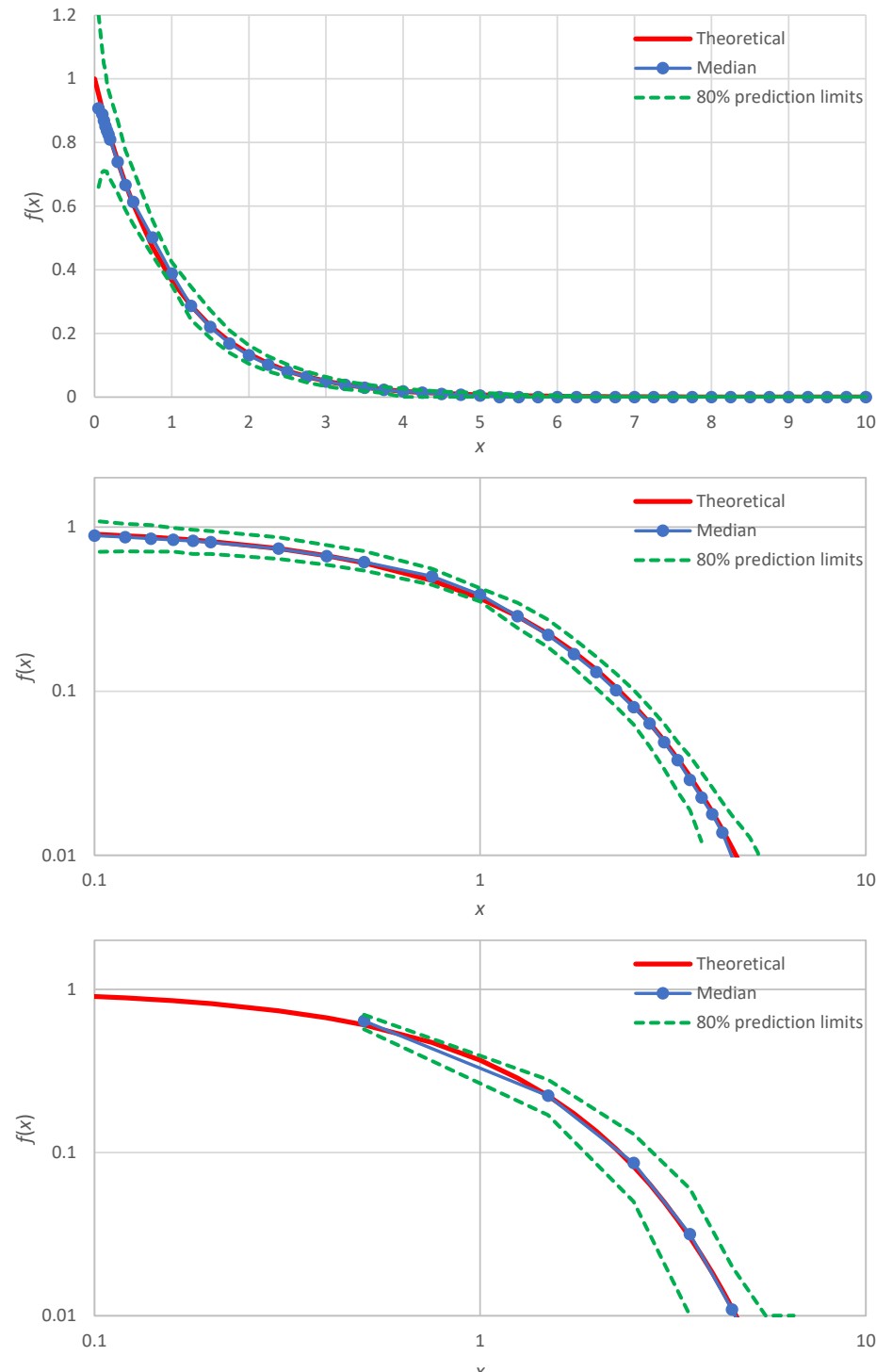

**Figure 2.** Illustration of the median estimate and uncertainty (in terms of prediction limits) of the probability density using the proposed method, plotted in Cartesian (**upper**) and logarithmic (**middle**) axes. The original results from the proposed method are interpolated at the points that are plotted in the graphs. For comparison, results for the classical histogram with 10 bins are also shown (**lower**), plotted with abscissae equal to the midpoints of the bins. The true distribution is exponential with parameters as in Table 4, from which 100 data series of $n = 100$ values each were generated and processed to produce the uncertainty band.

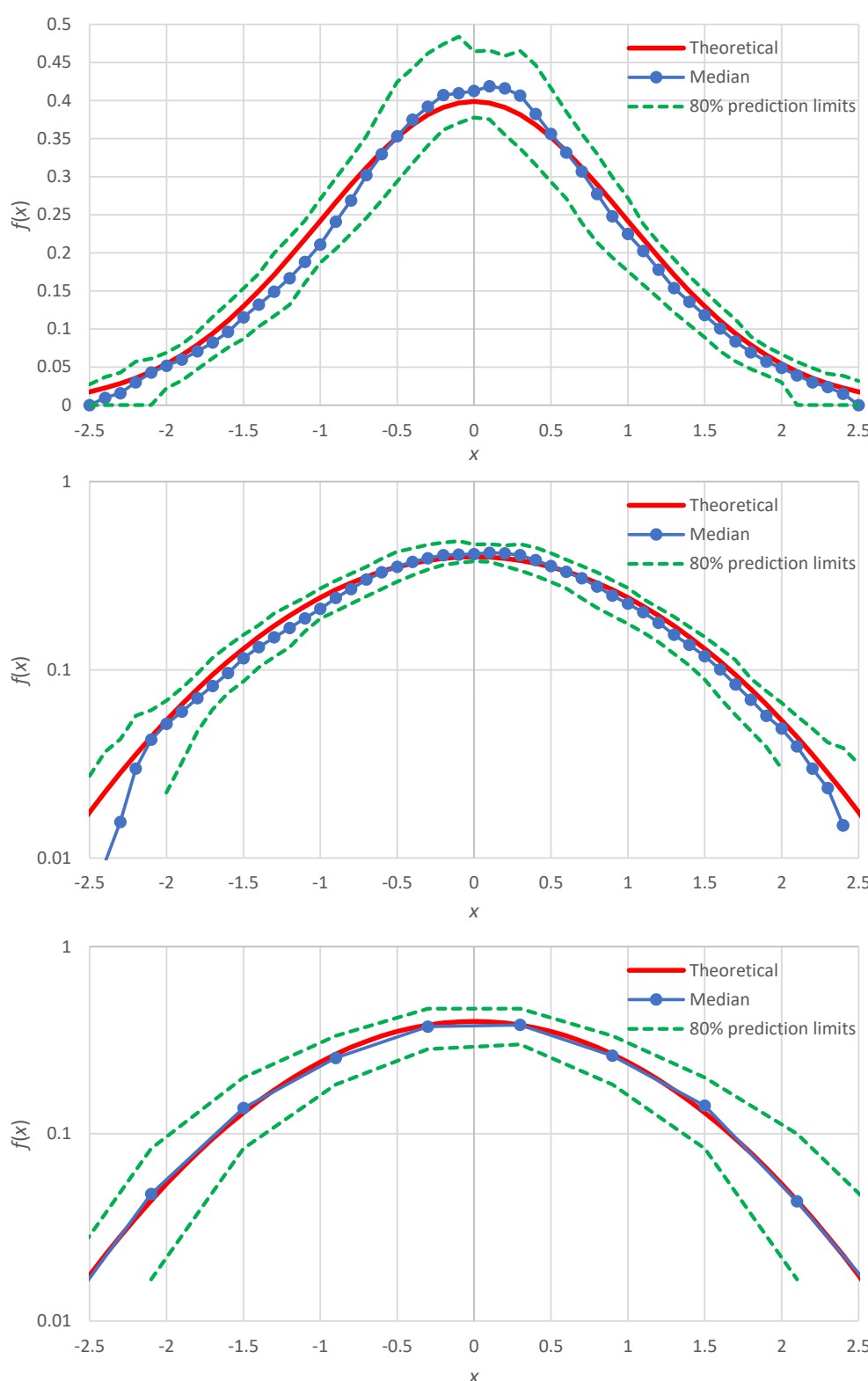

**Figure 3.** Illustration of the median estimate and uncertainty (in terms of prediction limits) of the probability density as in Figure 2 but for the normal distribution with parameters as in Table 4.

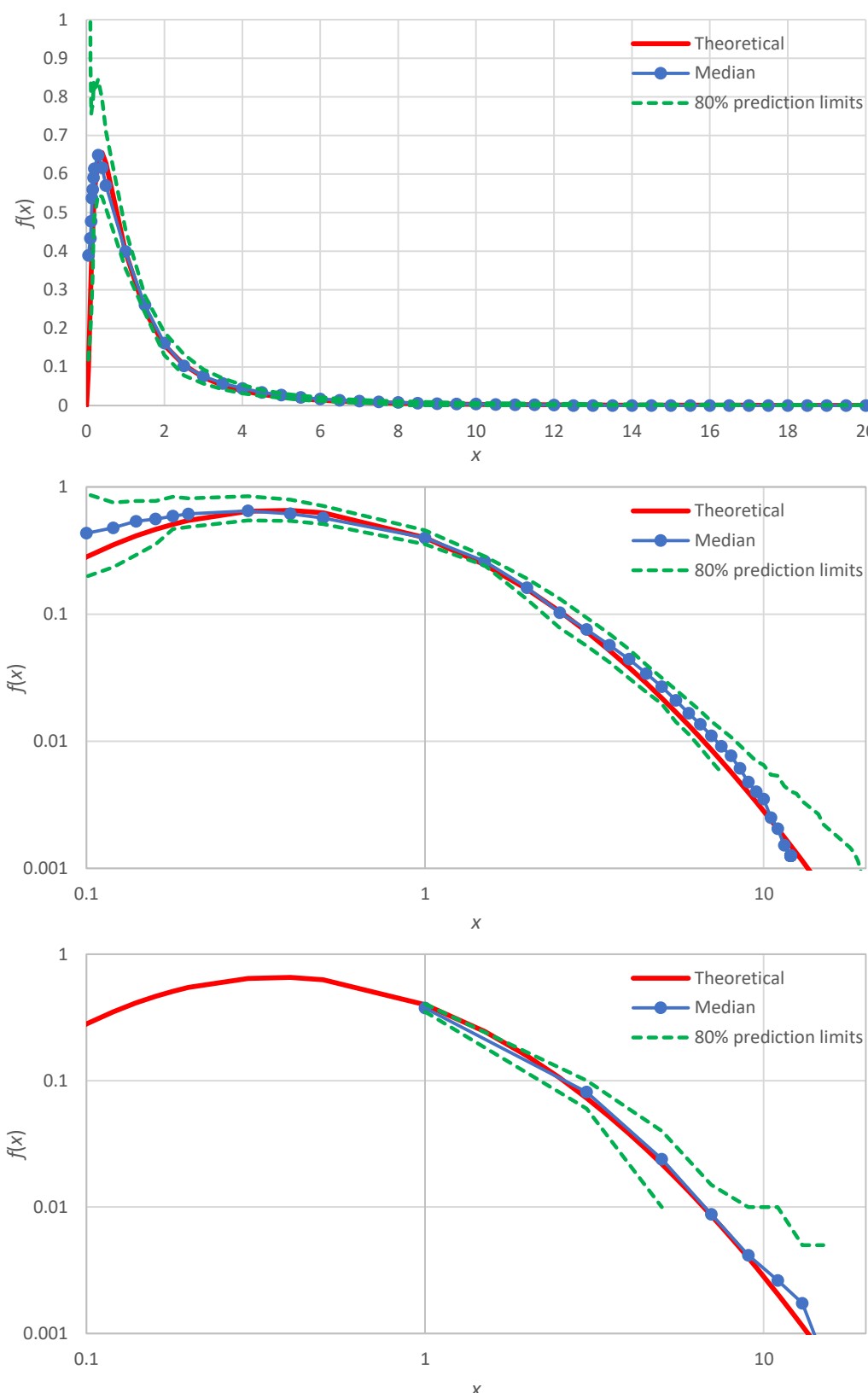

**Figure 4.** Illustration of the median estimate and uncertainty (in terms of prediction limits) of the probability density as in Figure 2 but for the lognormal distribution with parameters as in Table 4.

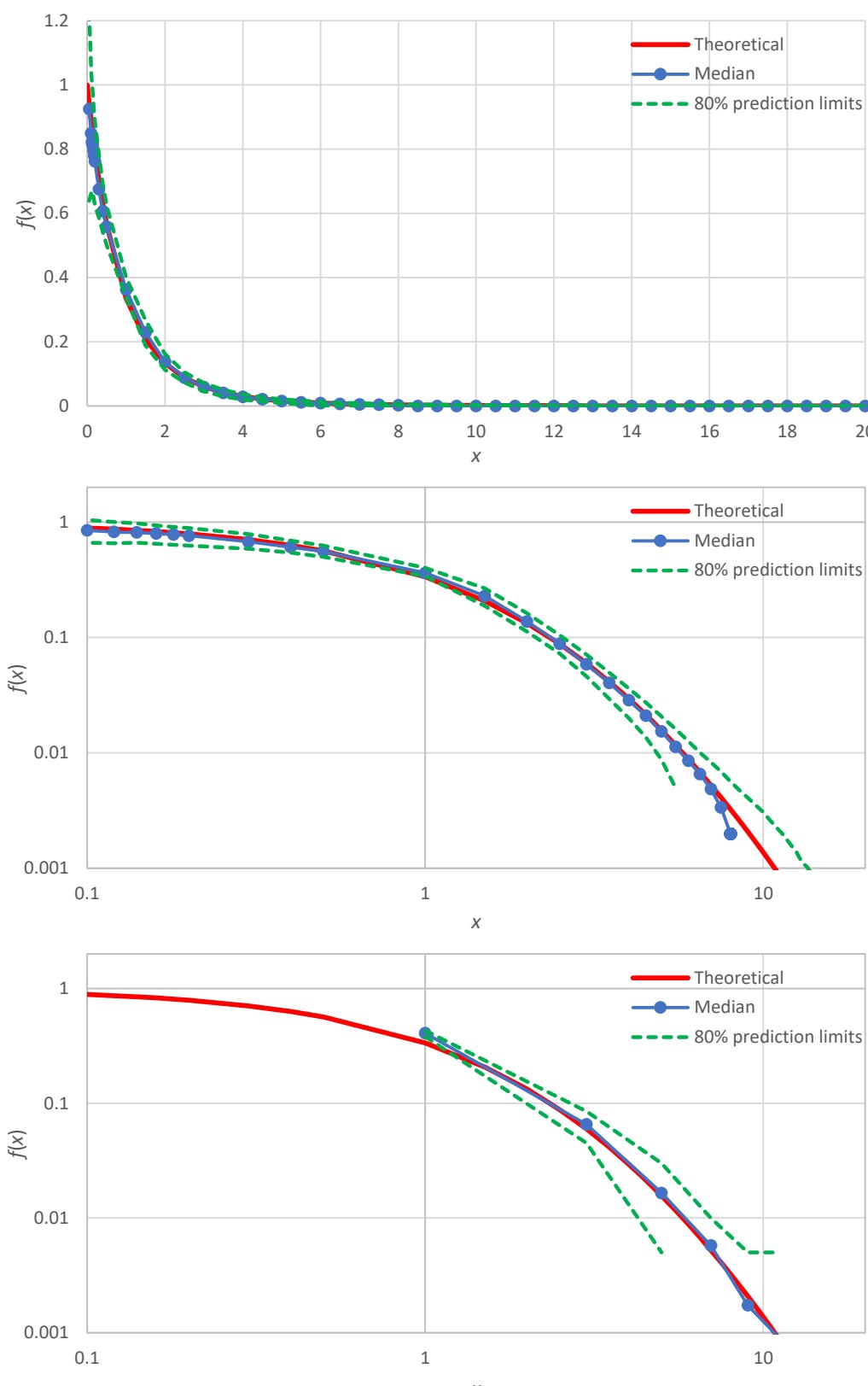

**Figure 5.** Illustration of the median estimate and uncertainty (in terms of prediction limits) of the probability density as in Figure 2 but for the Pareto distribution with parameters as in Table 4.

**Table 4.** Estimated means, standard deviations and entropies: averages from the 100 Monte Carlo simulations performed in the study with sample size 100.

| Distribution, Parameters | Mean | | | Standard Deviation | | | Entropy | |
|---|---|---|---|---|---|---|---|---|
| | True | Est. 1 * | Est. 2 * | True | Est. 1 | Est. 2 | True | Est. 2 |
| Exponential, $\mu = 1$ | 1 | 0.99 | 1.00 | 1 | 0.98 | 0.98 | 1 | 0.96 |
| Normal, $\mu = 0$, $\sigma = 1$ | 0 | 0.01 | 0.01 | 1 | 0.99 | 0.99 | 1.42 | 1.37 |
| Lognormal, $\varsigma = \lambda = 1$ | 1.65 | 1.69 | 1.80 | 2.16 | 2.26 | 2.31 | 1.42 | 1.42 |
| Pareto, $\xi = 0.2$, $\lambda = 1$ | 1.25 | 1.26 | 1.27 | 1.61 | 1.56 | 1.57 | 1.20 | 1.18 |

* Est. 1: Estimate from the typical sample statistics (without involving the probability density function); Est 2: Estimate from the empirical density function.

In all cases, the estimated probability density, expressed in terms of the median of the simulations (or the average thereof, which is very close to the median and was not plotted in the figures), harmonizes with the true shape of the probability density. This applies to the body of the distribution, as well as to the right and left tails, which are better discerned in the double logarithmic plots of the figures (middle panels). Slight discrepancies appear in the normal distribution (Figure 3), which will be discussed in Section 4. For comparison, in each of the figures, double logarithmic plots are also provided for the classical histograms (lower panels). In these, the simulated medians harmonize well with the true densities, yet the points are too few and the left tails of the distributions are not captured. As per the uncertainty, the proposed method clearly outperforms the histogram framework as the zones defined by prediction limits are quite narrower in the former case than in the latter.

*3.3. Entropy Estimation*

Most of the statistical estimators (e.g., of moments) do not involve the probability density and thus they do not require the estimation thereof. Entropy is an exception, because its very definition relies on the logarithm of the density function. Namely, the entropy $\Phi[\underline{x}]$ of the stochastic variable $\underline{x}$ is defined as:

$$\Phi[\underline{x}] := \mathrm{E}\left[-\ln \frac{f(\underline{x})}{\beta(\underline{x})}\right] = -\int_{-\infty}^{\infty} \ln \frac{f(x)}{\beta(x)} f(x)\mathrm{d}x \tag{34}$$

where $\beta(x)$ is a background measure, which can be any probability density, proper (with integral equal to 1) or improper (meaning that its integral diverges). Typically, it is an (improper) Lebesgue density, i.e., a constant with dimensions $[\beta(x)] = [f(x)] = [x^{-1}]$, so that the argument of the logarithm function be dimensionless. Here, we assume $\beta(x) = 1$.

The common practice is to estimate $f(x)$ through a histogram. However, this technique is not perfect because of the rough shape of the estimate and the subjective choices about the bins. These may result in a distorted estimate. To illustrate this, we consider the following example, based on the exponential distribution. For a sample of size $n$ from this distribution, the expected values of the highest of the $n$ values is $K'_n = \mu H_n$, where $H_n$ is the $n$th harmonic number, and that of the lowest is $\overline{K}'_n = \mu/n$ (see Koutsoyiannis [6], pp. 182–183). Therefore, the expected value of the range is $r := K'_n - \overline{K}'_n = \mu H_{n-1} \approx \mu(\gamma + \ln n)$. In the limiting case of only one bin ($k = 1$), the entropy will be estimated as $\hat{\Phi}_1[\underline{x}] = \ln r = \ln(\mu(\gamma + \ln n))$. At the opposite end, if we choose too many bins, $k > n$, so that each one contains either one or zero elements, then the probability density estimate will be either 0 (for the bins containing no element) or $1/(nw)$ where $w = r/k$. The former case does not contribute to entropy, so that the summation to estimate entropy (by converting the integral in (34)) is made on the bins containing elements. The entropy in this case will be $\hat{\Phi}_k[\underline{x}] = -\sum_i \ln \hat{f}_i \hat{f}_i w =$ $\sum_i \ln(nw)(1/nw)w = \ln(nw) = \ln(nr/k) = \ln r + \ln(n/k)$. This depends on $k$ and by choosing a large $k$ it can become arbitrarily small (even negative) as $\ln(n/k) < 0$. The true entropy is $\Phi[\underline{x}] = \ln e\mu$ (Table 2). Now, assuming, for instance, $\mu = 1, n = 100, k = 500$ the

true entropy is $\Phi[\underline{x}] = 1$, while the estimates are $\hat{\Phi}_1[\underline{x}] = 1.65 > 1$ and $\hat{\Phi}_{500}[\underline{x}] = 0.04 < 1$. These values indicate the subjectivity of the estimates. A good choice of $k$ will result in good estimate, yet we cannot be certain about its reliability as we cannot control the factors producing the estimation errors.

The more detailed representation of the probability density allows for a better estimation of entropy. However, we must have in mind that in unbounded stochastic variables there is uncertainty beyond the maximum (or the minimum) observed value in the sample, which we denote as $c$. For a variable bounded by zero from below and unbounded from above, we can proceed with the correction proposed by Koutsoyiannis and Sargentis [19] to take into account the contribution of probability density for $x > c$.

Specifically, the expectation of any function $g(\underline{x})$ can be calculated as

$$\mathrm{E}[g(\underline{x})] := \int_0^\infty g(x)f(x)\mathrm{d}x = A^g + B^g, A^g := \int_0^c g(x)f(x)\mathrm{d}x, B^g := \int_c^\infty g(x)f(x)\mathrm{d}x \quad (35)$$

The quantity $A^g$ can directly be estimated from the available data, by approximating the integral with a sum. Assuming that the data are given in terms of density estimates $\hat{f}_i$ at points $x_i$, with $i = 1, \dots m$ and $x_m \equiv c$, we have:

$$\hat{A}^g = \sum_{i=1}^n g\left(\frac{x_{i-1} + x_i}{2}\right)\hat{f}_i\,(x_i - x_{i-1}) = \sum_{i=1}^n g\left(\frac{x_{i-1} + x_i}{2}\right)(\hat{F}_i - \hat{F}_{i-1}) \quad (36)$$

To estimate the quantity $B^g$ we assume that beyond $c$ an exponential approximation is sufficient for the purpose:

$$f(x) = \mathrm{e}^{-x/\kappa+\beta} \Rightarrow \overline{F}(x) = \kappa\mathrm{e}^{-x/\kappa+\beta} = \kappa f(x), x \geq c \quad (37)$$

where $\beta$ and $\kappa$ are parameters to be estimated. For the moment of order $p$ of the distribution we have:

$$B_p := \int_c^\infty x^p \mathrm{e}^{-x/\kappa+\beta}\,\mathrm{d}x = \mathrm{e}^\beta\,\Gamma\left(p+1, \frac{c}{\kappa}\right)\kappa^{p+1} \quad (38)$$

where $\Gamma(a, x) := \int_x^\infty t^{a-1}\mathrm{e}^{-t}\mathrm{d}t$ is the incomplete gamma function. In particular for $p = 0, 1, 2$ we have

$$B_0 = \overline{F}(c), B_1 = B_0(c+\kappa), B_2 = B_0\left(\kappa^2 + (c+\kappa)^2\right) \quad (39)$$

with

$$\kappa = \frac{\overline{F}(c)}{f(c)} \quad (40)$$

Furthermore, for the entropy we have

$$B^\Phi = B_0(1 - \ln f(c)) = B_0(1 - \ln B_0 + \ln \kappa) \quad (41)$$

We assume that $B_0 = \overline{F}(c)$ is known from the data, estimated as $\hat{B}_0 = \hat{\overline{F}}(c)$. In the case that, in addition, $f(c)$ can reliably be estimated as $\hat{f}(c)$, the sought parameter $\kappa$ is estimated as $\hat{\kappa} = \hat{\overline{F}}(c)/\hat{f}(c)$. However, in the outmost available point of the tail, $f(c)$ is not adequately reliable. An alternative option is to estimate from the data the quantity $B_2$

$$\hat{B}_2 = \hat{\mu}_2' - \hat{A}_2 \quad (42)$$

where $\hat{\mu}_2'$ is determined by the standard sample estimator. In this case, solving the rightmost of Equation (39) for $\kappa$ we find

$$\hat{\kappa} = \frac{1}{2}\left(\sqrt{\frac{\hat{B}_2}{\hat{B}_0} - c^2} - c\right) \tag{43}$$

This allows estimation of $B^g$ for the expectation of any function $g(\underline{x})$. We note that the correction is applied only if the quantity within the square root turns out to be positive (which is usually the case).

If the distribution is unbounded from both below and above (e.g., in the normal distribution), then we apply the procedure twice for the lower and upper tail. In this case, we have $B_{0_L} = F(c_L)$ and $B_{0_R} = \overline{F}(c_R)$ where the subscripts L and R refer to left (below) and right (above), respectively. In this case, Equation (42) automatically includes both corrections and in Equation (43) we should replace $\hat{B}_0$ with the sum $\hat{B}_{0L} + \hat{B}_{0R}$.

Application of this technique for the Monte Carlo simulations described above are given in Table 4, in which we see that the method works well.

## 4. Discussion

The method is described above in its minimal configuration. Improvements are possible in several ways, e.g., to take into account possible dependence of the consecutive variables (i.e., when we do not have an observed sample but a time series), or to use more accurate representations of the relationship between K-moments and their corresponding values of distribution functions. These issues are studied in Koutsoyiannis [6] (pp. 147–249)—but not for the density estimation.

Here, we discuss the case of improvement of the distribution function estimation for the normal distribution, for which the simulation results (Figure 3) showed slight discrepancies. As shown in [6] (pp. 209–213), the more accurate representation of the $\Lambda$-coefficients for the normal distribution are

$$\Lambda_p \approx \Lambda_\infty + \frac{A}{p} - B\ln\left(1 + \frac{1}{\ln(p+1)}\right), A = \Lambda_1 - \Lambda_\infty + B\ln\left(1 + \frac{1}{\ln 2}\right) \tag{44}$$

$$\overline{\Lambda}_p \approx \overline{\Lambda}_\infty + \frac{\overline{A}}{p} + \overline{B}\ln\left(1 + \frac{1}{\ln(p+1)}\right), \overline{A} = \overline{\Lambda}_1 - \overline{\Lambda}_\infty - \overline{B}\ln\left(1 + \frac{1}{\ln 2}\right) \tag{45}$$

where for the normal distribution $B = \overline{B} = 0.73$. In this case, we find:

$$F\left(K_p'\right) = 1 - \frac{1}{\left(\Lambda_\infty + B\ln\left(1 + \frac{1}{\ln(p+1)}\right)\right)p + \Lambda_1 - \Lambda_\infty - B\ln\left(1 + \frac{1}{\ln 2}\right)} \tag{46}$$

$$F\left(\overline{K}_p'\right) = \frac{1}{\left(\overline{\Lambda}_\infty - \overline{B}\ln\left(1 + \frac{1}{\ln(p+1)}\right)\right)p + \overline{\Lambda}_1 - \overline{\Lambda}_\infty + \overline{B}\ln\left(1 + \frac{1}{\ln 2}\right)} \tag{47}$$

In addition, noticing that the points plotted around $p = 1$ are at greater distances to each other than the other points, we can use non-integer values of $p$ between 1 and 2. Repeating the Monte Carlo simulation with these two modifications, we get the results shown in Figure 6 (lower panel), which are in better agreement with the true density than those of the minimal version, also reproduced in Figure 6 (upper panel).

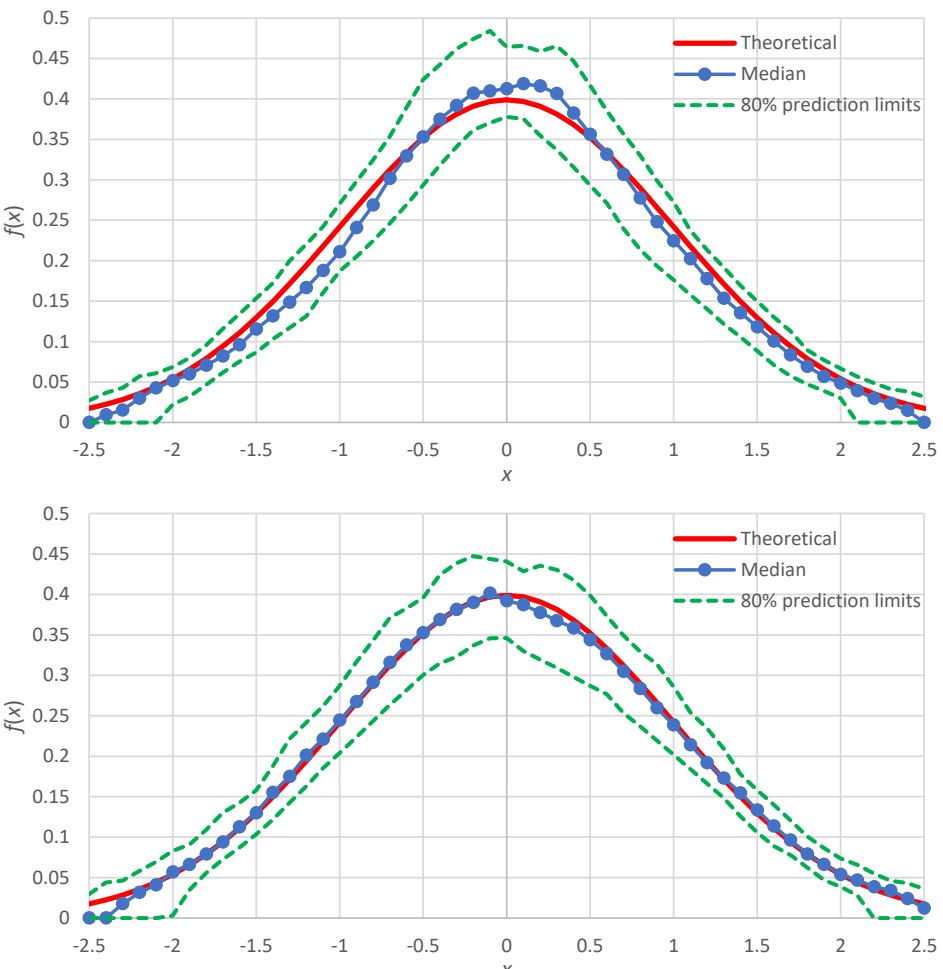

**Figure 6.** Comparison of the Monte Carlo simulation results for the normal distribution (with parameters as in Table 4) with the minimal version of the method (**upper**; copy of the upper panel of Figure 3) and the enhanced version (**lower**; using Equations (46) and (47)).

Even if we use the minimal version, it is possible and sometimes useful to calculate $\hat{f}\left(\hat{K}'_p\right)$ for non-integer order $p$. In this case, we can even find (see derivation in Appendix B) an analytical expression for the probability density estimate, which for the noncentral K-moments is

$$\hat{f}\left(\hat{K}'_p\right) = \frac{\Lambda_\infty \, \hat{\bar{F}}\left(\hat{K}'_p\right)^2}{\sum_{i=\lceil p\rceil}^{n} b_{inp} c_{inp} x_{(i:n)}} \tag{48}$$

where

$$\hat{K}'_p = \sum_{i=\lceil p\rceil}^{n} b_{inp}\, x_{(i:n)}, \quad \hat{\bar{F}}\left(\hat{K}'_p\right) = \frac{1}{\Lambda_\infty p + \Lambda_1 - \Lambda_\infty}, \quad c_{inp} := \frac{1}{p} + H_{i-p} - H_{n-p} \tag{49}$$

and $b_{inp}$ is given by Equation (18). It is reminded that $H_n$ denotes the $n$th harmonic number. Similar equations can be developed for the tail K-moments.

Notice in Equations (48) and (49) that the lower limit of the sum is not the non-integer $p$ but its floor $\lceil p\rceil$. This means that the functions $\hat{F}\left(\hat{K}'_p\right)$ and $\hat{f}\left(\hat{K}'_p\right)$ will not be fully continuous, but only left continuous. However, the discontinuities are practically negligible for $p < n - 1$.

A final note is that the proposed method, in addition to providing point estimates $\hat{f}\left(\hat{K}'_p\right)$, can also produce interval estimates of $f\left(\hat{K}'_p\right)$ by means of confidence limits deter-

mined by Monte Carlo simulation. These differ from the prediction limits of Section 3.2, which were constructed for a known true distribution function that was used to generate several (in our case 100) realizations of samples. For the confidence limits, the true distribution is assumed unknown and the simulation is made from the estimate $\hat{F}(x)$, which is determined at the points $\hat{K}'_p$ and $\hat{\bar{K}}'_p$. The generation from this distribution is easy: the values are generated as $x = \hat{F}^{-1}(u)$, where $u$ is a random number from the uniform distribution in [0, 1]. The inverse function $\hat{F}^{-1}$ is determined by interpolation (and occasionally extrapolation) from the sequence of points $(x_i, \hat{F}(x_i))$, with the sequence of $x_i$ being $\left( \hat{\bar{K}}'_n, \hat{\bar{K}}'_{n-1}, \dots, \hat{\bar{K}}'_1 \equiv \hat{K}'_1, \ \hat{K}'_2, \dots \hat{K}'_{n-1}, \hat{K}'_n \right)$. The interpolation is better made in terms of the quantity $\ln(\hat{F}/(1-\hat{F}))$ instead of $\hat{F}$.

An illustration for the example of Figure 1 (lognormal distribution) is shown in Figure 7 (lower panel), where the produced confidence band is also compared with uncertainty band defined by the prediction limits of the lognormal distribution (upper panel of Figure 7). We observe that: (a) the confidence band (lower panel) has about the same width as the uncertainty band (upper panel), and (b) the true distribution, which was not used in the Monte Carlo simulation of the lower panel, is contained within the produced confidence band—as it should.

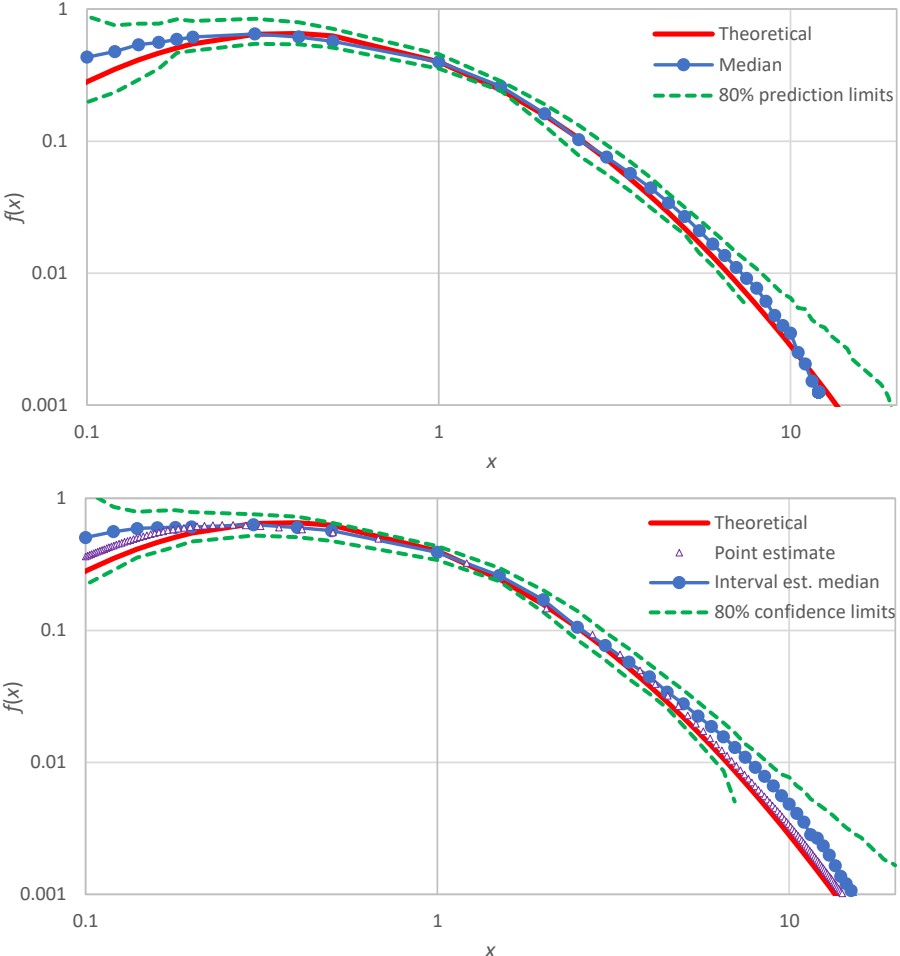

**Figure 7.** Comparison of the Monte Carlo simulation results for prediction limits of the lognormal distribution (with parameters as in Table 4) with the minimal version of the method (**upper**; copy of the middle panel of Figure 4) and for the confidence limits of the empirical probability density of Figure 1 (**lower**). The plotted "point estimates" are precisely those shown in Figure 1.

## 5. Conclusions

For continuous stochastic variables, the proposed framework is an improved, detailed and smooth alternative to the widely used concept of the histogram, which provides only a gross representation of the probability density. The proposed framework is based on the concept of knowable moments (K-moments) and its characteristics that make possible the estimate of the probability density are:

- The ability to reliably estimate from a sample, moments of high order, up to the sample size.
- The ability to assign values of the distribution function to each estimated value of K-moment.
- The smoothness of the estimated values, which are linear combinations of a number of observations, rather than based on a single observation as in other approaches.

The latter characteristic is crucial in estimating the probability density, which is the first derivative of the distribution function.

Prominent characteristics of the proposed method of density estimation, confirmed by several applications of the method for a variety of distribution functions, are:

- The faithful representation of the true density, both in the body and the tails of the distribution.
- The dense and smooth shape, owing to the ability to estimate values of the density at very many points (even for any arbitrary point) within the range of the available observations.
- The low uncertainty of estimates.
- The ability to provide both point and interval estimates (confidence limits), with the latter becoming possible by Monte Carlo simulation.
- The simplicity of the calculations, which can be made in a typical spreadsheet environment.

Specifically, the calculations include the following steps, which summarize the numerical part of the proposed method.

1. We sort the observed sample in ascending order.
2. We calculate the estimates $\underline{\hat{K}}'_p$, $\overline{\hat{K}}'_p$, $i = 1, \ldots, n$ from Equations (17) and (18).
3. We estimate the coefficient $\Lambda_1$ from Equation (24) and calculate $\overline{\Lambda}_1$ from Equation (30).
4. We assume the tail indices of the distribution and choose values of $\Lambda_\infty$ and $\overline{\Lambda}_\infty$ from Table 3 (see [6] (pp. 209–213), for additional distributions).
5. We calculate the estimates $\hat{F}\left(\hat{K}'_p\right)$ and $\hat{F}\left(\overline{\hat{K}}'_p\right)$ from Equations (28) and (32) for all $\underline{\hat{K}}'_p$, $\overline{\hat{K}}'_p$ derived in step 2. (By plotting the tails $\underline{\hat{\overline{F}}}\left(\hat{K}'_p\right)$ and $\hat{F}\left(\overline{\hat{K}}'_p\right)$ in double logarithmic graphs, we check whether the empirically estimated tail indices agree with those assumed in Step 4 and, if not, we repeat steps 4 and 5 with new estimates).
6. We calculate the estimates of $f(x)$ from Equation (33).

These calculations are illustrated in the Supporting Information, which contains a spreadsheet accompanying this paper. This gives a full-scale application of the method, related to the construction of Figure 1.

A problem of the method is that it requires one to have an idea of the type of the true distribution, in terms of its upper- and lower-tail indices, in order to estimate the asymptotic $\Lambda$-coefficients ($\Lambda_\infty, \overline{\Lambda}_\infty$). If the sample size is large, the K-moments can support the estimation of these tail indices (cf. points steps 4 and 5 above). However, for small samples ($n < \sim 100$) their estimation becomes problematic and higher uncertainty is induced. This issue requires further investigation.

**Supplementary Materials:** As a supporting information, an Excel spreadsheet with application of the method (including the construction of Figure 1) can be downloaded at: https://www.mdpi.com/article/10.3390/sci4040050/s1 or from http://www.itia.ntua.gr/2256/.

**Funding:** This research received no external funding but was conducted for scientific curiosity.

**Institutional Review Board Statement:** Not applicable.

**Informed Consent Statement:** Not applicable.

**Data Availability Statement:** Not applicable.

**Acknowledgments:** I thank three reviewers for their constructive comments which helped improve the paper.

**Conflicts of Interest:** The author declares no conflict of interest.

## Appendix A. Illustration of Alternative Techniques

As stated in the Introduction, if we use the empirical estimate of the distribution function $F(x)$ by Equation (1), we get a staircase-like function, which is illustrated in the upper panel of Figure A1 using the same data as in Figure 1. This is not a continuous function and thus it does not allow calculation of the derivative, which is the probability density $f(x)$. One may think of keeping only the left-hand point in each stair step (the triangles in Figure A1) and replace the staircase form with a broken line (drawing straight-line segments between two consecutive points depicted as triangles). However, even in this case there will be roughness, which is hugely magnified if we take the derivative (slope of each linear segment). This is depicted in the lower panel of Figure A1, where the thus estimated values of $f(x)$ vary by several orders of magnitude and the different point estimates are far distant from the true $f(x)$. The huge roughness and variability that appear exclude the possibility of regarding the numerical results by this method as estimates of the probability density.

As an alternative that can derive a smooth density estimate, the use of kernel functions has been proposed, as described in the Introduction. The kernel estimate of probability density is derived as

$$\hat{f}(x,h) = \frac{1}{nh}\sum_{i=1}^{n} K\left(\frac{x - x_i}{h}\right) \tag{A1}$$

where $K(x)$ is the kernel function, which has the property $\int_{\infty}^{\infty} K(x)\mathrm{d}x = 1$, and $h > 0$ is a parameter, termed the bandwidth. Two of the most common kernel functions are the uniform:

$$K(x) = \begin{cases} 1/2, & |x| \leq 1 \\ 0, & |x| > 1 \end{cases} \tag{A2}$$

and the normal (Gaussian):

$$K(x) = \frac{e^{-x^2/2}}{\sqrt{2\pi}} \tag{A3}$$

These are illustrated in Figure A2, where we may observe that, despite producing a smooth curve, the entire shape remains erratic and follows the empirical histogram, rather than approaching the true density. In addition, the curves are much more subjective than the histogram per se, as they depend on the user choices of the kernel and its bandwidth.

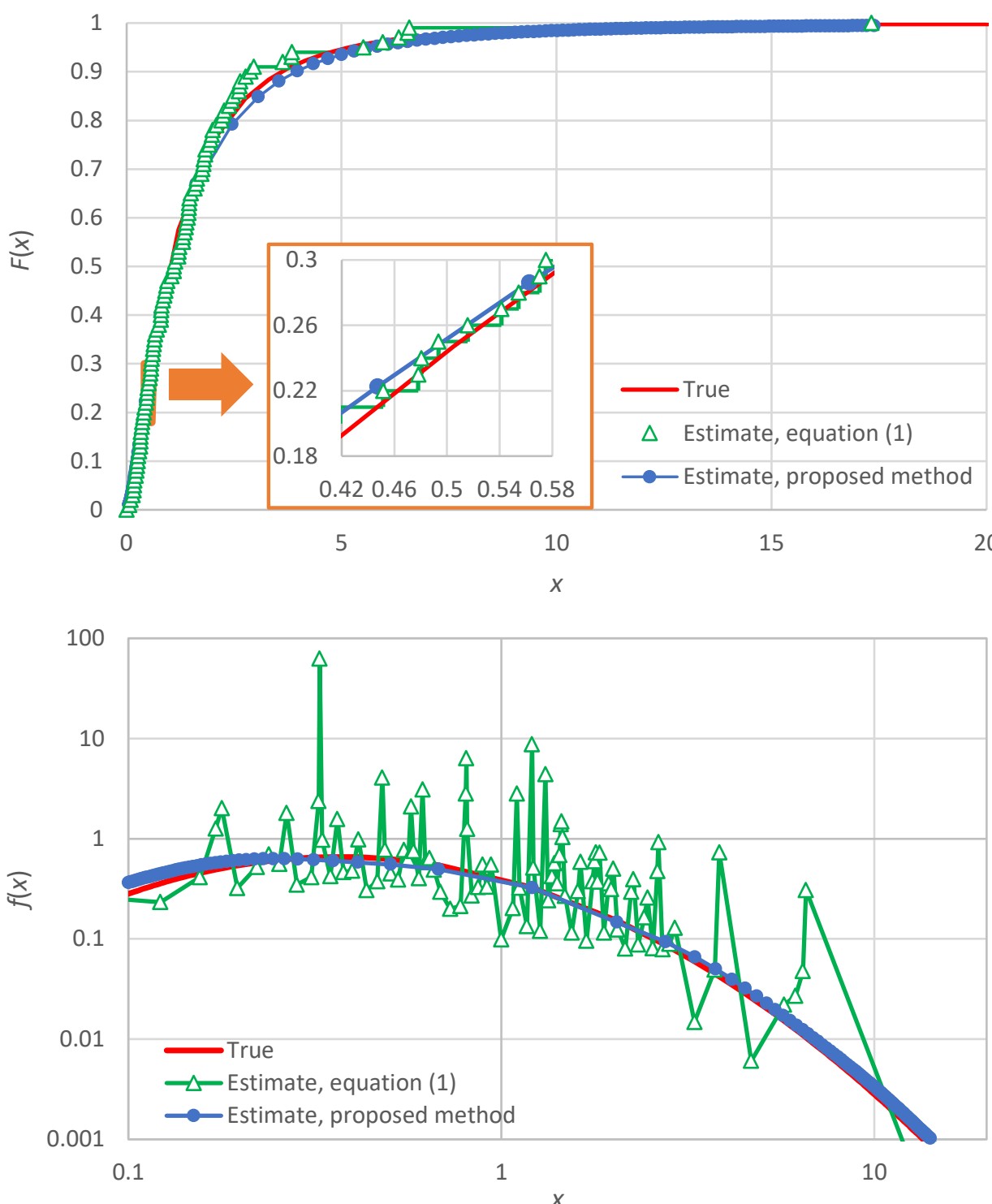

**Figure A1.** Illustration of the estimate of the distribution function $F(x)$ using Equation (1) (**upper**, with inset to better depict the staircase form) and that of the probability density $f(x)$ (**lower**) as the numerical derivative of $F(x)$ with the staircase form replaced by a broken line form. The data series of Figure 1 was used ($n = 100$ values, generated from a lognormal distribution with parameters $\varsigma = \lambda = 1$; see Table 2). The abscissae of the points of estimates are the midpoints of the intervals $\left(x_{(i:n)}, x_{(i+1:n)}\right)$. As the estimates of $f(x)$ vary by orders of magnitude, logarithmic axes are used. In both panels the estimates by the proposed method (from Figure 1) are also shown for comparison.

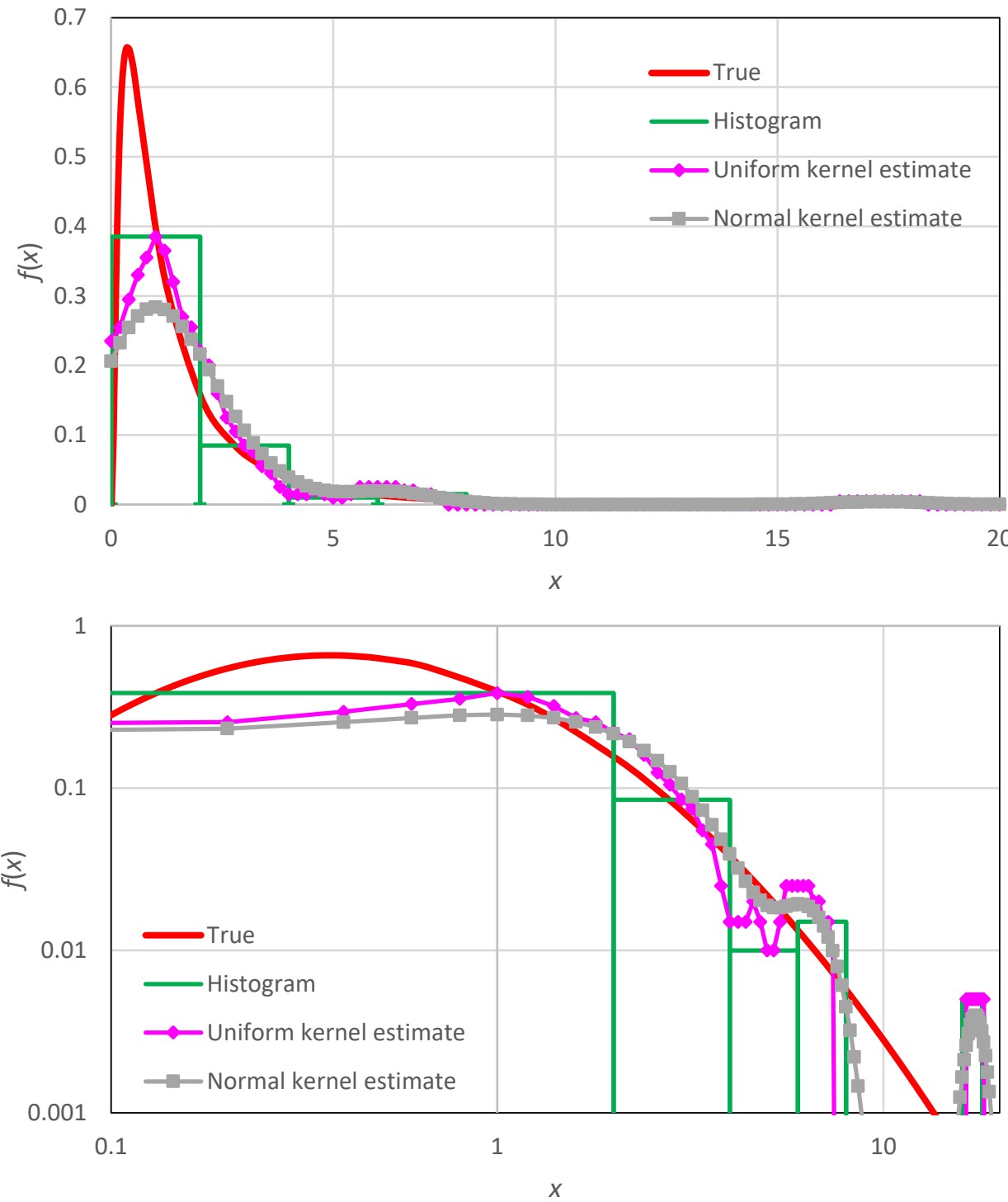

**Figure A2.** Illustration of the probability density estimate using (a) the histogram (Equation (4) with 10 bins for the range [0, 20]), (b) the uniform kernel (Equation (A2) with $h = 1$) and (c) the normal kernel (Equation (A3) with $h = 1$), plotted in Cartesian (**upper**) and logarithmic (**lower**) axes. The data series of Figure 1 was used ($n = 100$ values, generated from a lognormal distribution with parameters $\varsigma = \lambda = 1$; see Table 2).

**Appendix B. Proof of Equation (48)**

We start from the obvious relationship

$$\frac{\mathrm{d}F\left(K'_p\right)}{\mathrm{d}p} = \frac{\mathrm{d}F\left(K'_p\right)}{\mathrm{d}K'_p}\,\frac{\mathrm{d}K'_p}{\mathrm{d}p} \tag{A4}$$

where the first term in the right-hand side is the density $f\left(K'_p\right)$. The left-hand side can be approximated using Equation (28), from which we find

$$\frac{\mathrm{d}F\left(K'_p\right)}{\mathrm{d}p} = \frac{\Lambda_\infty}{\left(\Lambda_\infty p + \Lambda_1 - \Lambda_\infty\right)^2} \tag{A5}$$

The second term in the right-hand side of Equation (A4) can be approximated by using the estimate of $K'_p$. Combining Equations (17) and (18) and replacing $p$ with the floor $p$ in the lower limit of the sum, we find:

$$\hat{K}'_p = \sum_{i=\lceil p \rceil}^{n} \frac{p}{n}\,\frac{\Gamma(n-p+1)}{\Gamma(n)}\,\frac{\Gamma(i)}{\Gamma(i-p+1)}\,x_{(i:n)} \tag{A6}$$

By taking the derivative with respect to $p$, after the algebraic manipulations we get:

$$\frac{\mathrm{d}\hat{K}'_p}{\mathrm{d}p} = \sum_{i=\lceil p \rceil}^{n} \frac{p}{n}\,\frac{\Gamma(n-p+1)}{\Gamma(n)}\,\frac{\Gamma(i)}{\Gamma(i-p+1)}\left(\frac{1}{p} + H_{i-p} - H_{n-p}\right)x_{(i:n)} \tag{A7}$$

or

$$\frac{d\hat{K}'_p}{dp} = \sum_{i=\lceil p \rceil}^{n} b_{inp}c_{inp}x_{(i:n)} \tag{A8}$$

where we have used the definitions of $b_{inp}$ and $c_{inp}$ in Equations (18) and (49), respectively. By combining all above we find:

$$\hat{f}\left(\hat{K}'_p\right) = \frac{\Lambda_\infty}{\left(\Lambda_\infty p + \Lambda_1 - \Lambda_\infty\right)^2}\,\frac{1}{\sum_{i=\lceil p \rceil}^{n} b_{inp}c_{inp}x_{(i:n)}} \tag{A9}$$

which can also be written in the form of Equation (48).

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
