# Peer review of "Replacing Histogram with Smooth Empirical Probability Density Function Estimated by K-Moments"

_sci, doi:10.3390/sci4040050_

Round 1

Reviewer 1 Report

1) It is beneficial to improve the paper with more comprehensive review of different traditional and new kernels, for example, Atomic Kernels from Atomic Machine Learning. Current 11 references seem not enough.

2) Also, illustrating the problem with a picture at the start would improve the overall attractiveness of the paper for non-in-depth-specialists. 

3) The conclusion 'the proposed framework can replace the widely-used concept of the histogram' seems too ambitious. Would it be better to 'improve' rather than 'replace'?

Author Response

The replies are contained in the attached pdf file.

Author Response

(The authors gave the same response as above.)

Author Response

(The authors gave the same response as above.)

Round 2

Reviewer 3 Report

I have read with attention the reply from the author to the referees and the corresponding changes in the revised version of Replacing histogram with smooth empirical probability density function estimated by K-moments 

I am happy with both the responses and the changes. 

I am happy to recommend publication of the manuscript in Sci.